# Radiation Hormesis in Barley Manifests as Changes in Growth Dynamics Coordinated with the Expression of *PM19L-like*, *CML31-like*, and *AOS2-like*

**DOI:** 10.3390/ijms25020974

**Published:** 2024-01-12

**Authors:** Elizaveta Kazakova, Irina Gorbatova, Anastasia Khanova, Ekaterina Shesterikova, Ivan Pishenin, Alexandr Prazyan, Mikhail Podlutskii, Yana Blinova, Sofia Bitarishvili, Ekaterina Bondarenko, Alena Smirnova, Maria Lychenkova, Vladimir Bondarenko, Marina Korol, Daria Babina, Ekaterina Makarenko, Polina Volkova

**Affiliations:** 1Laboratory of Molecular and Cellular Radiobiology, Russian Institute of Radiology and Agroecology of National Research Centre “Kurchatov Institute”, 249035 Obninsk, Russia; elisabethafeb19@gmail.com (E.K.); gorbatova.irina.96@mail.ru (I.G.); micenyk-anastasi@mail.ru (A.K.); eshesterikova89@gmail.com (E.S.); pishenin.ivan@gmail.com (I.P.); prazyan22@gmail.com (A.P.); yana.manuhina@yandex.ru (Y.B.); bitarishvili.s@gmail.com (S.B.); bondarenco.e@gmail.com (E.B.); sas.smirnova@mail.ru (A.S.); lychenkovamariya@gmail.com (M.L.); bvs79@mail.ru (V.B.); podobedmyu@gmail.com (M.K.); babinadd@gmail.com (D.B.);; 2Independent Researcher, 2440 Geel, Belgium

**Keywords:** abscisic acid, calcium signalling, eustress, growth stimulation, *Hordeum vulgare*, γ-irradiation, jasmonic acid, low doses, promotors

## Abstract

The stimulation of growth and development of crops using ionising radiation (radiation hormesis) has been reported by many research groups. However, specific genes contributing to the radiation stimulation of plant growth are largely unknown. In this work, we studied the impact of the low-dose γ-irradiation of barley seeds on the growth dynamics and gene expression of eight barley cultivars in a greenhouse experiment. Our findings confirmed that candidate genes of the radiation growth stimulation, previously established in barley seedlings (*PM19L-like*, *CML31-like*, and *AOS2-like*), are significant in radiation hormesis throughout ontogeny. In γ-stimulated cultivars, the expression of these genes was aligned with the growth dynamics, yield parameters, and physiological conditions of plants. We identified contrasting cultivars for future gene editing and found that the γ-stimulated cultivar possessed some specific abiotic stress-responsive elements in the promotors of candidate genes, possibly revealing a new level of radiation hormesis effect execution. These results can be used in creating new productive barley cultivars, ecological toxicology of radionuclides, and eustress biology studies.

## 1. Introduction

This decade is a critical time to take action to mitigate the worst impacts of climate change on people and ecosystems [1]. The increasing impact of climate change on agricultural systems determines the need to grow crops that can withstand harsh environmental conditions such as heat, drought, salt stress, flooding, and disease outbreaks [2]. These stressors affect plant fitness and, ultimately, food production. However, low doses or intensities of stress exposure can be beneficial for plant growth and yield [3,4]. Such an effect is known as hormesis or eustress, a dose–response phenomenon of growth stimulation after the application of low doses of adverse factors. In contrast, high doses of these factors induce growth inhibition or can even be lethal [5,6]. Hormetic effects can expand the ability of researchers to mitigate the adverse outcomes of climate change on crop production [7].

Among physical stress factors, pre-sowing low-dose ionising radiation is a well-known growth-promoting factor, increasing plant size and biomass [8,9,10]. However, the agronomic practice of seed irradiation is difficult to implement due to technical restrictions and instability outside the controlled conditions [11]. Therefore, identifying molecular pathways of radiation growth stimulation coupled with gene editing technologies may be a better approach to creating high-yielding and stress-tolerant crop cultivars. Functional genomics of plant molecular responses to stress, such as ionising radiation, can provide insights into candidate stress tolerance genes and help reveal the main players in radiation stimulation, which can be further used to recreate the stimulating effect without irradiation through gene editing technologies [12]. Additionally, ionising radiation as a provoking genotoxic factor becomes a promising tool for searching for multiple stress tolerance candidate genes [13].

Common barley, *Hordeum vulgare* L., is one of the most important crops with a high nutritional value and adaptability, although its maximum productivity in the world has declined due to climate change [14,15,16,17]. Barley cultivars readily respond to low-dose seed radiation treatment [18], and we revealed several candidate metabolites and genes as possible players in the radiation hormesis effect in this crop [11,19,20,21]. Changes in the expression of rice homologues *PM19L*, *CML31*, and *AOS2* in seedlings of barley cultivars with different sensitivities to irradiation were earlier proposed as possible determinants of radiation hormesis [21]. The *CML31-like* is involved in calcium signal transduction, *AOS2-like* participates in jasmonate signalling, and *PM19L-like*—in abscisic acid signalling [22,23,24].

Recent genetic engineering and gene editing breakthroughs have expanded the possibilities for creating new plant cultivars, offering opportunities to improve agronomically relevant environmental tolerance traits in cultivated species [25]. For barley, genomic tools, such as a recently updated reference genome assembly [26] and CRISPR-Cas protocols for precise gene editing [27,28,29], are available. Still, applying these tools for crop improvement requires the knowledge of candidate genes of tolerance and genomic composition of target cultivars.

This study aimed to further confirm barley *PM19L-like*, *CML31-like*, and *AOS2-like* genes as players in radiation stimulation and targets for gene editing. Their expression was studied through different ontogeny stages in eight barley cultivars after γ-irradiation of seeds, and various growth, physiological, and yield parameters were assessed. The structure of candidate genes in the two most promising cultivars was studied for subsequent gene editing. Here, we report first, to our knowledge, molecular targets for precise gene editing based on radiation hormesis effect mechanisms and discuss a possible role of *cis*-regulatory elements in radiation hormesis execution.

## 2. Results

### 2.1. Dynamics of Phenological Stages of Barley Grown from Irradiated and Control Seeds in a Greenhouse Experiment

Eight cultivars of barley with different sensitivities to γ-irradiation were used in the greenhouse experiment: γ-stimulated Fox 1, Ratnik, Eryoma, “no morphological effect” Grees, Timofey, Vivat, Fedos, and the γ-inhibited cultivar Leon. The radiation sensitivity assessments were performed earlier based on morphological reactions of 7-day-old barley seedlings to seed irradiation at the dose of 20 Gy [21]. The dynamics of ontogenetic stages were assessed in plants growing from seeds irradiated at 20 Gy and in control plants.

For irradiated and control plants of all cultivars, the change in micro-phenological stages was monitored using a scale suggested in [30], from 00 to 99. The stages on this scale are determined visually through phenological features of organ formation: germination (00–09); seedling growth (10–19); tillering (20–29); stem elongation (30–39); booting (40–49); inflorescence (ear/panicle) emergence or earing (50–59); flowering (60–69); and three stages of grain ripening: milk grain (70–79), dough grain (80–89), and fully ripe (90–99). Plant tissues were sampled at seedling, tillering, booting, and earing stages to assess the expression of candidate radiation hormesis genes *PM19L-like*, *CML31-like*, and *AOS2-like*. At the booting stage, chlorophyll fluorescence was recorded, and yield parameters were evaluated at the end of the experiment (Figure 1).

Figure 2 summarises dynamics in phenological stages of γ-stimulated and γ-inhibited cultivars grown from irradiated and control seeds in a greenhouse experiment. Vertical stripes in Figure 2 indicate the stages at which differences in the ontogenetic dynamics were revealed between irradiated and control plants.

According to the ranking of seedling responses to radiation, the Fox 1 cultivar was identified as the most γ-stimulated [21]. In the current work, the dose of 20 Gy stimulated the dynamics of Fox 1 ontogenesis at many phenological stages. Plants grown from irradiated seeds at stages 13 and 14 developed two days faster than the control; at stages 15 and 17—five and six days, respectively; at stage 18, irradiated plants moved to the next stage three days earlier than the control. The tillering stage started in the irradiated plants five days earlier than in the control; however, the flowering phase began two days later than in the control plants. Therefore, the dynamics of the ontogenetic stages of the Fox 1 cultivar indicated that plants grown from 20 Gy-irradiated seeds began the transition to development stages 13, 14, 15, and 17 (seedling growth) and tillering earlier than the control (Figure 2a), confirming the Fox 1 cultivar as γ-stimulated [21].

For another γ-stimulated cultivar, Ratnik, the plants grown from irradiated seeds finished growth stages 10 and 11 one day later than the control, while stage 15 ended twelve days earlier than in the control (some plants in the control never moved to stage 16), and stage 16 began four days earlier than in the control plants. The control plants did not reach stage 18, while the irradiated plants did. The booting and flowering stages began three and four days later in plants grown from irradiated seeds (Figure 2b). Such shifts in development dynamics may indicate a greater biomass gain by irradiated plants. Indeed, when assessing the yield parameters, we observed a significant increase in the weight of straw of the irradiated plants of the Ratnik cultivar compared to the control (*p* = 0.0006) and in the number of stems (*p* = 0.031, Appendix A).

For the γ-stimulated cultivar Eryoma, in irradiated plants, early ontogenetic stages 8 and 9 ended, and stages 11, 12, and 14 began one to two days later than in the control. At the same time, the irradiated plants moved to stage 17 four days earlier compared to the control, while the transition to stage 18 was again delayed. The transitions to subsequent macro stages were synchronous in all plants (Figure 2c).

In the γ-inhibited cultivar Leon in the greenhouse experiment, almost all stages for the irradiated and control plants began on the same day. Stages 11–16 were completed earlier in plants grown from irradiated seeds than in the control (Figure 2d). At the same time, at the booting stage, in contrast to the γ-stimulated cultivars, an increase in some photosynthesis parameters, PhiNO, v_H_^+^, and PS1, was noted compared with the control (*p* = 0.002, *p* = 0.005, and *p* = 0.044, respectively, Appendix A).

Figure 3 summarises dynamics in phenological stages of the “no morphological effect” cultivars grown from irradiated and control seeds in a greenhouse experiment. In the Grees cultivar, most phenological stages in the plants grown from irradiated seeds either began or ended earlier than the control (by two to seven days, Figure 3a). However, tillering in plants grown from irradiated seeds was delayed (at the same time, tillering shoots were not developed in all plants). Based on yield estimates, control plants had a larger average stem height per plant (*p* = 0.0006) and spike weight (*p* = 0.009) compared to the irradiated plants (Appendix A). The linear electron flow (LEF) parameter of photosynthesis was increased in the control plants (Appendix A).

For the cultivar Timofey, growth stages 10, 12, 13, 14, and 18 began several days earlier in irradiated plants compared to control ones, and stage 19 was recorded seven days earlier; flowering also started several days earlier. At the same time, stages 16 and 17 in the irradiated plants began with a noticeable lag, and the start of the elongation, booting, and earing stages did not differ between conditions (Figure 3b).

In the cultivar Vivat, stage 9 started earlier and ended later in the irradiated plants. Stages 13 and 14 in the irradiated plants began two and four days earlier, respectively. From stage 15 to stage 19, the transition to subsequent developmental stages was faster in the control plants; by stage 19, the lag in the irradiated plants reached 10 days. The tillering phase in the irradiated barley began on the 39th day after planting (DAP), and in the control plants, it started on the 43rd DAP. The stages of elongation, booting, earing, and flowering occurred simultaneously in both conditions (Figure 3c). For this cultivar, an increase in plant height (*p* = 0.038), average height of stems per plant (*p* = 0.0003), and number of stems (*p* = 0.019) was noted in the irradiated barley compared with the non-irradiated (Appendix A).

The simultaneous onset of the elongation, booting, earing, and flowering in stages 14–17 was noted for the Fedos cultivar (Figure 3d). Stages 8 and 10 began earlier in the control, and stage 18 started in the control plants eight days faster than in the irradiated ones. This cultivar was second to last in the ranking by the severity of the effect of seed gamma irradiation on seedlings [21]. At the booting stage, the coefficient of photochemical fluorescence quenching qL significantly increased in the irradiated plants compared to the control (*p* = 0.004, Appendix A), while the g_H_^+^ values were lower than in the control plants (*p* = 0.008, Appendix A).

On the 90th DAP, all cultivars had a seed ripening stage of “milk grain”—77. On the 99th DAP, plants with ears had ripeness stages from 77 to 85. After 2.5 weeks, the barley seeds reached full ripeness and were harvested.

Thus, the γ-irradiation of barley seeds at 20 Gy can change the dynamics of ontogenesis. Compared to radiosensitivity assessments at early stages of growth (up to seven days after germination [21]), more barley cultivars positively reacted to pre-sowing seed irradiation at later stages of ontogeny, expanding possible targets for further gene editing. 

### 2.2. Analysis of Differentially Expressed Genes in the Roots and Leaves of Irradiated Plants

The expression of candidate genes for radiation hormesis, homologues of rice *PM19L*, *CML31*, and *AOS2* was modulated at different ontogenetic stages of barley plants in response to the γ-irradiation of seeds. Those changes in expression depended on the stage of development and the cultivar being studied (Figure 4, Figure 5 and Figure 6).

#### 2.2.1. Differential Expression of *PM19L-like*, *CML31-like*, and *AOS2-like* Genes in γ-Stimulated Cultivars

*PM19L-like* encodes the membrane protein PM19L, which is involved in plant responses to abiotic stress through ABA-dependent signalling [23,31,32]. In roots of irradiated seedlings of γ-stimulated cultivars Fox 1 and Ratnik and leaves of the Eryoma cultivar, the downregulation of *PM19L-like* was found. In contrast, in seedling leaves of the Ratnik cultivar, this gene was upregulated (Figure 4). At the tillering stage of development, the upregulation of *PM19L-like* was detected in the leaves of the Fox 1 cultivar. During booting of the Ratnik cultivar, *PM19L-like* was downregulated (Figure 4).

Calcium ion (Ca^2+^) is an intracellular second messenger involved in many signal transduction pathways in plants. Calmodulin-like (CML) proteins are primary calcium sensors in plant growth and stress responses [22,33,34]. *CML31* gene encodes in rice a probable calcium-binding protein CML31. The expression of *CML31-like* in leaves of the Fox 1 cultivar was not detected at some ontogenetic stages. It was downregulated in seedling roots of γ-stimulated cultivars Fox 1 and Ratnik, while in leaves of Ratnik seedlings, it was detected only in the control plants (Figure 4). At the earing stage, *CML31-like* was upregulated in leaves of the Ratnik cultivar; in leaves of the Erema cultivar, the gene was expressed only in irradiated plants (Figure 4).

**Figure 4 ijms-25-00974-f004:**
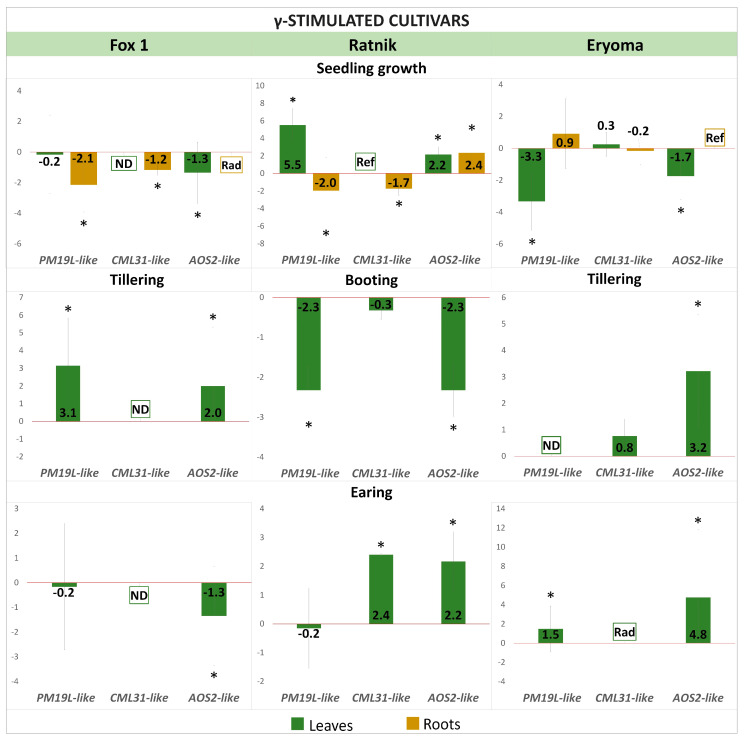
Expression of candidate genes (log_2_FC) in leaves and roots at different stages of ontogenesis of γ-stimulated barley cultivars grown from γ-irradiated seeds. Values > 0—upregulation, values < 0—downregulation. The graphs reflect log-normalised mean relative expression ± SE (N = 2–3). *—log_2_FC ≥ |1|. Ref—expression was revealed in non-irradiated samples only; Rad—expression was revealed in irradiated samples only; ND—no expression detected.

In plants, the hormone jasmonic acid (JA) and its derivatives regulate responses to biotic and abiotic stressors [35,36]. Rice *AOS2*, a homologue of *AT5G42650* in *Arabidopsis thaliana*, is a member of the cytochrome P450 family and functions as an allene oxide synthase. This enzyme is involved in the pathway associated with fatty acid biosynthesis and catalyses the dehydration of hydroperoxide to unstable allene oxide [35]. The expression of *AOS2-like* was increased in leaves and roots of seedlings of the γ-stimulated cultivar Ratnik. At the same time, it was downregulated in the leaves of the seedlings of γ-stimulated cultivars Fox 1 and Eryoma. In roots of Fox 1 seedlings, the expression of *AOS2-like* was recorded only in plants after irradiation, in contrast to the Eryoma cultivar, where expression was observed only in control plants (Figure 4). At the developmental stage of tillering, the expression of *AOS2-like* in leaves changed in the opposite direction to the seedling stage. At the booting stage, *AOS2-like* expression decreased in the Ratnik cultivar. At the earing stage, *AOS2-like* expression again changed to the opposite one compared to the previous stage of development in the Fox and Ratnik cultivars. On leaves of the Eryoma cultivar, *AOS2-like* expression remained upregulated.

#### 2.2.2. Differential Expression of *PM19L-like*, *CML31-like*, and *AOS2-like* Genes in “No Effect” Cultivars

At the seedling stage, the upregulation of *PM19L-like* was recorded in leaves and roots of the Grees cultivar and the roots of the Timofey and Fedos cultivars. The upregulation persisted in the Grees cultivar at booting and earing stages (Figure 5). At the earing stage, the Timofey cultivar was characterised by downregulation of *PM19L-like*, while at the booting stage, the expression of this gene was not detected. For the Fedos cultivar at the booting and earing stages, the expression was detected only in control plants (Figure 5).

The expression of *CML31-like* decreased in the leaves of seedlings of the cultivars Timofey and Fedos and increased in roots of Timofey seedlings. For the Vivat cultivar, expression in the leaves of seedlings was detected only in plants grown from irradiated seeds (Figure 5). At the booting stage, the leaves of cultivars Grees, Fedos, and Timofey were characterised by the differential expression of *CML31-like*. In the earing stage in the leaves, upregulation was observed for the Grees cultivar, while expression was recorded only for control plants in the Timofey and Fedos cultivars (Figure 5).

**Figure 5 ijms-25-00974-f005:**
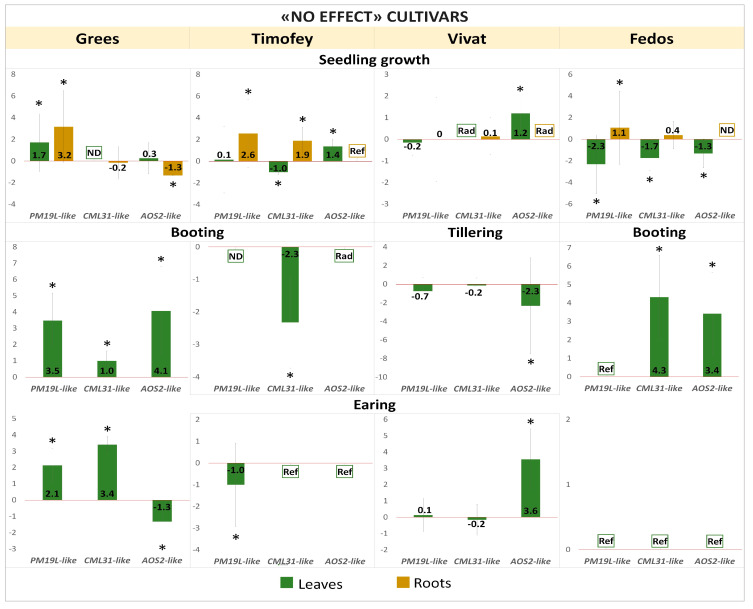
Expression of candidate genes (log_2_FC) in leaves and roots at different stages of ontogenesis of “no effect” cultivars of barley grown from γ-irradiated seeds. Values > 0—upregulation, values < 0—downregulation. The graphs reflect log-normalised mean relative expression ± SE (N = 2–3). *—log_2_FC ≥ |1|. Ref—expression was revealed in non-irradiated samples only; Rad—expression was revealed in irradiated samples only; ND—no expression detected.

The expression of the *AOS2-like* gene was increased in the leaves of Timofey and Vivat seedlings and decreased in the leaves of the Grees cultivar. At the same time, this gene was downregulated in the roots of Grees seedlings. In roots of Timofey seedlings, expression was observed only in the control, and for Vivat—only in roots of plants after irradiation of seeds (Figure 5). A change in the expression pattern of the *AOS2-like* to the opposite direction compared with the seedling stage was noted for the Vivat cultivar at the tillering stage (downregulation) and the Fedos cultivar at the booting stage (upregulation). At the booting stage, *AOS2-like* expression increased in leaves of the Grees cultivar, while for the Timofey cultivar it was noted only for irradiated plants (Figure 5). At the earing stage, the expression changed again to the opposite direction from the previous stage for the cultivars Grees and Vivat, and for the cultivars Timofey and Fedos, it was recorded only in control plants (Figure 5).

#### 2.2.3. Differential Expression of *PM19L-like*, *CML31-like*, and *AOS2-like* Genes in the γ-Inhibited Cultivar “Leon”

In leaves of the γ-inhibited cultivar Leon, the upregulation of *PM19L-like* was recorded (Figure 6). The expression of *CML31-like* was not detected in roots and leaves of Leon seedlings and was strongly downregulated in leaves at the earing stage (Figure 6). At the booting stage, *AOS2-like* was upregulated in leaves. It should be noted that the expression of *AOS2-like* in the leaves of the Leon cultivar changed only at the later stages of development: booting (upregulation in leaves) and earing (downregulation in leaves) in contrast to other cultivars, where the modulation of *AOS2-like* expression was already observed at the seedling stage. Interestingly, in previous work on seedlings under controlled conditions [21], we observed a change in the expression of *AOS2-like* only in γ-stimulated cultivars.

**Figure 6 ijms-25-00974-f006:**
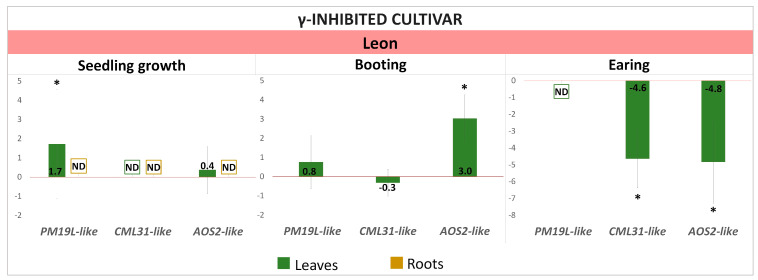
Expression of candidate genes (log_2_FC) in leaves and roots at different stages of ontogenesis of γ-inhibited Leon cultivar of barley grown from γ-irradiated seeds. Values > 0—upregulation, values < 0—downregulation. The graphs reflect log-normalised mean relative expression ± SE (N = 2–3). *—log_2_FC ≥ |1|. ND—no expression detected.

### 2.3. Summary of Seed Irradiation Effects on Growth Dynamics and Gene Expression

Changes in ontogenesis stages, along with the differential expression of candidate genes, are shown in Appendix A. In leaves of the γ-stimulated cultivar Fox 1 at the tillering stage, the upregulation of the *PM19L-like* and *AOS2-like* genes was accompanied by the onset of the tillering stage earlier than in non-irradiated plants (Appendix A). For the γ-stimulated Ratnik cultivar, we observed the upregulation of *PM19L-like* at the seedling stage in leaves and *AOS2-like* in leaves and roots. At the same time, *CML31-like* and *PM19L-like* were downregulated in roots. These expression changes accompanied a later end of stages 10–11 and an early onset of stage 15 compared to non-irradiated plants (Appendix A). The reduced expression of *PM19L-like* and *AOS2-like* in the leaves of the Ratnik cultivar may be associated with a delay of the booting stage in irradiated plants (Appendix A).

The earing stage for the irradiated plants of the Grees cultivar began earlier than in the control, and at this stage, the upregulation of *PM19L-like* and downregulation of *AOS2-like* were noted in the leaves (Appendix A). In irradiated seedlings of the Timofey cultivar, stages 10–14 and 18–19 started earlier than the control ones and were accompanied by the upregulation of *PM19L-like* and *CML31-like* in seedling roots and by the downregulation of *CML31-like* and the upregulation of *AOS2-like* in seedling leaves (Appendix A). For the Vivat cultivar, the earlier beginning of the tillering stage in plants after irradiation was accompanied by the downregulation of *AOS2-like* (Appendix A). The analysis of the expression of candidate genes at different ontogeny stages of irradiated plants is summarised in Figure 7.

Based on assessments of the expression of candidate radiation hormesis genes and data on ontogenetic dynamics, chlorophyll fluorescence, and yield, we identified the most contrasting cultivars for future genetic editing of barley: the γ-stimulated cultivar Ratnik and the γ-inhibited (in current study—γ-indifferent) cultivar Leon. Sequencing was carried out for these cultivars, and the structure of rice *CML31*, *AOS2*, and *PM19L* homologues was studied.

### 2.4. CML31, AOS2, and PM19L Rice Homologues Gene Structure in Ratnik and Leon Barley Cultivars

#### 2.4.1. Gene Structure of *CML31-like* in Ratnik and Leon Cultivars

The alignment of the sequenced fragments of *CML31-like* in the Ratnik cultivar to the Morex reference sequence (*HORVU.MOREX.r3.3HG0322130*) included a 2047 bp promoter region, 668 bp coding region, and 592 bp terminator region. Three small insertions of 7, 6, and 5 bp were observed in the promoter region, two 3 bp insertions in the coding sequence, and one 8 bp insertion in the 3′-untranslated region (Figure 8a).

For the Leon cultivar, the reference sequence included 2115 bp promoter region, 651 bp coding region, and 605 bp terminator region. In the promoter region, three insertions of 40, 44, and 11 bp were revealed. In the coding sequence, one single nucleotide insertion was identified (Figure 8b).

DOFCOREZM and SV40COREENHAN motives were revealed in the promoter region of the Leon cultivar, with insertions 44 and 12 bp long (positions 876–919 and 570–581 bp, correspondingly, upstream of the start codon). The Ratnik cultivar, instead, possesses the MYBCORE binding site that is absent in Leon because of the 12 bp insertion in position 570–581 bp.

#### 2.4.2. Gene Structure of *AOS2-like* in Ratnik and Leon Cultivars

The alignment of the sequenced fragments of *AOS2-like* in the Ratnik cultivar to the Morex sequence (*HORVU.MOREX.r2.4HG0328590*) included 2043 bp of promoter, 1552 bp of coding region, and 5684 bp of terminator part. A large insertion of 37 bp was revealed in the promoter, and one single-nucleotide insertion and three deletions of 1 bp were discovered in the coding region. Nine 5–19 bp indels were identified throughout the terminator region (Figure 9a).

The alignment of *AOS2-like* in the Leon cultivar included 2084 bp of promoter, 1574 bp of coding fragment, and 5654 bp of terminator sections. One deletion of 5 bp and one insertion of 16 bp were detected in the promoter region; multiple single-nucleotide insertions and deletions and one 7 bp insertion were observed throughout the coding sequence, and two insertions of 9 and 23 bp, as well as a deletion of 8 bp, were identified in terminator part (Figure 9b).

Unlike the Ratnik cultivar, Leon presented CAATBOX1 motif in position 569–609 bp (16 bp insertion) of the *AOS2-like* promoter, AACACOREOSGLUB1 binding site in position 603–604 bp, and CACTFTPPCA1 in 681–697 bp.

#### 2.4.3. Gene Structure of *PM19L-like* in Ratnik and Leon Cultivars

The alignment of the sequenced fragments of the *PM19L-like* in the Ratnik cultivar to the reference sequence (*HORVU.MOREX.r3.5HG0537660*) included 2134 bp of promoter, 941 bp of coding fragment, and 695 bp of terminator. The promoter revealed three deletions of 8, 12, and 19 bp and three large insertions of 19, 25, and 71 bp. The terminator contained two insertions of 8 and 29 bp and one 15 bp deletion (Figure 10a).

For *PM19L-like* in the Leon cultivar, the structure included 2019 bp of promoter, 939 bp of coding fragment, and 698 bp of terminator sections. Three insertions of 5, 7, and 29 bp, as well as a 19 bp deletion, were revealed in the terminator region (Figure 10b).

A comparative analysis of the promoter region revealed sequences in the Leon cultivar that are absent in the Ratnik cultivar: 551–558 and 587–598 bp upstream of the start codon. These regions contain binding sites for transcription factors DOFCOREZM and POLLEN1LELAT52 (587–598 bp) and CANBNNAPA and RAV1AAT (position 551–558 bp). Due to deletions of 8 and 12 bp, the Ratnik cultivar gained binding sites for PRECONSCRHSP70A and MYBCORE, which are absent in the Leon cultivar.

For the promoter regions of the studied genes, conservative regulatory motifs, such as TATA box, AT-TATA box, and CAAT box, and a number of *cis*-regulatory elements involved in response to various stress factors were found and characterised (Appendix A). Among them, in particular, were *cis*-elements ABRE, W-box, DRE, MBS, MYB motifs, STRE-motif, G-box, and AE-box (Appendix A).

## 3. Discussion

During several years of rigorous research, we studied physiological and molecular mechanisms of radiation hormesis in barley plants [18,19,21,37] and used various omics approaches to find candidate molecules of radiation stimulation [11,20]. The validation of those findings in the large greenhouse experiment makes us one step closer to the successful application of radiation tools in genetic technologies for agricultural needs.

### 3.1. Differential Expression of Candidate Genes and Ontogenetic Dynamics 

The irradiation of seeds influenced seedlings’ development rate in the green house, promoting a faster transition between stages of plant development, even for cultivars, which were considered not reacting to radiation exposure in previous small-scale experiments. The up- or downregulation of *PM19L-like* in leaves of irradiated plants were associated with promoting or suppressing growth dynamics (cultivars Fox 1, Ratnik, Grees, and Fedos). The opposite pattern was observed for *PM19L-like* expression in roots for cultivars Fox 1, Ratnik, and Fedos. Although a clear connection of *CML31-like* expression with growth dynamics has not been revealed for leaves or roots, the most γ-responsive cultivars Fox 1 and Ratnik showed similar patterns of expression, where lower expression was associated with faster growth rates. *AOS2-like* upregulation seems to be associated with the more rapid development of γ-stimulated cultivars (Figure 7).

The homologue of *PM19L* in *A. thaliana*, *AT1G04560*, belongs to the family of AWPM-19-like proteins, which was first reported in wheat and is associated with the regulation of embryo development and dormancy and is involved in response to various abiotic stress factors [23,32,38,39]. In rice, the *OsPM19L1* gene responded to drought, high salinity, and low temperature stresses through stress-induced ABA signalling pathways [32]. It has been shown that the upregulation of the *PM19L* homologue can be caused by an increase in the ABA level [32]. Genes of this family can directly participate in ABA transport and provide drought resistance when overexpressed [40]. The *PM19L* gene, probably a membrane-localised protein in the ABA signalling pathway, plays a key role in ensuring the drought resistance of *Paeonia lactiflora* [41]. The pattern of *PM19L-like* regulation, along with the ontogenetic dynamics, suggests that the accumulation of *PM19L-like* transcripts in leaves rather than roots is accompanied by enhanced growth rates. Since an enhanced expression of its homologues is associated with improved antioxidant activity and better photosynthetic performance [41], it is plausible to consider that *PM19L-like* expression in leaves is essential for the radiation hormesis phenomenon. The downregulation of this gene in Fedos seedlings (Appendix A) was accompanied by a decrease in thylakoid proton conductivity g_H_^+^ (Appendix A), reflecting a slowing of the ATP synthase through “metabolism-related” regulation [42] and an increase in qL reflecting the quinone Q_A_ redox state [42].

Allenoxide synthase (AOS) is the second enzyme in the biosynthesis of the plant hormone jasmonic acid [24,43]. AOS plays a decisive role in JA-dependent stress responses and developmental processes [24]. JA is known to interact with ABA to modulate plant response and tolerance to abiotic stresses. JA promotes ABA accumulation, activating the expression of ABA biosynthesis genes [44]. At the early growth stages, patterns of expression of *AOS2-like* and *PM19L-like* are similar in studied cultivars (Figure 7), pointing to the possible role of the ABA-JA crosstalk in the growth-promoting or growth-retarding effect of ionising radiation.

A family of CML proteins is involved in the transduction of Ca^2+^ signals during adaptation to abiotic stress [45]. The modulation of the expression of the *CML31* homologue in γ-stimulated cultivars may indicate an important role of calcium signalling during the establishment of the effect of radiation hormesis (Appendix A). However, apparently, shifts in the dynamics of growth phases were associated with the expression of the *PM19L-like* and *AOS2-like* to a greater extent.

The patterns of expression of three genes in γ-stimulated cultivars indicate a possible role of JA, ABA, and calcium signalling in the radiation stimulation effect, especially at the early growth stages. These three genes were suitable for further gene editions in target barley cultivars. Special attention was given to the significant stimulation of yield parameters while identifying the most prominent cultivar, as accelerated growth dynamics alone may not reflect the final biomass accumulation. After taking into account the promotive shifts in ontogenetic stages (Figure 2 and Figure 7), increased dry weight of stroke (+42%, Appendix A), and stable photosynthetic parameters (Appendix A), we selected the γ-stimulated cultivar Ratnik as the most promising for further genetic editing to increase barley stress tolerance and productivity. The Leon cultivar was the least responsive regarding development rate (Figure 7), and its photosynthetic parameters were significantly deteriorated (Appendix A, an increased PhiNO parameter reflecting the ratio of incoming light lost via non-regulated processes). This cultivar was chosen as the most contrasting to γ-stimulated cultivars.

### 3.2. CML31-like, AOS2-like, and PM19L-like Gene Structure in Ratnik and Leon Cultivars

We sequenced three barley homologues of rice genes *PM19L*, *CML31*, and *AOS2* to evaluate the structure of target genes and identify insertions and deletions in Ratnik and Leon cultivars in comparison with the reference sequences of *H. vulgare* to analyse *cis*-regulatory elements in the promoter regions and transcription factor binding sites (Figure 8, Figure 9, Figure 10 and Appendix A), and to receive a better understanding of target genes’ regulation.

In the promoter regions of the studied genes, we found conserved regulatory motifs TATA-box [46], AT-ТАТА-box and CAAT-box [46,47], and stress-sensitive *cis*-regulatory elements (Appendix A), indicating that the proteins coded by these genes are involved in response to various biotic and abiotic factors. Indeed, ABRE is a *cis*-sequence sensitive to ABA, which plays a significant role in protecting plants from abiotic stress; W-box is a *cis*-sequence recognised by the WRKY family of transcription factors and is associated with the response to biotic and abiotic stress; DRE—an element that responds to water deficiency; MBS—drought; MYB motifs are dehydration-sensitive elements that are involved in responses to various stressors, in particular, in response to drought; STRE-motif—stress-responsive element; G-box and AE-box—participation in the reaction to light [31,47].

Promoter variations can change expression levels [48,49,50], and some *cis*-or *trans*-regulatory functions involved in abiotic stress tolerance responses may be lost or acquired due to natural gene variations [51]. Our data show the emergence of distinct motifs in the promoter regions of sequenced genes (Figure 8, Figure 9 and Figure 10). These motifs, being targets of transcription factors, may influence the regulation of the gene expression of Ratnik and Leon cultivars. For example, the *CML31-like* gene is expressed in control seedlings of the Ratnik cultivar (Figure 4) but not in the Leon cultivar (Figure 6). The promoter region of *CML31-like* in Ratnik has an MYBCORE binding site for plant transcription factors MYB, which affects tolerance to abiotic stress [52], and this motif is absent in the Leon cultivar due to a 12 bp insertion. This insertion may be responsible for the lack of transcription of the *CML31* gene homologue and lower tolerance of seedlings to irradiation stress. Similar situations have been described in the literature: a 28 bp deletion was identified in *Brassica rapa* upstream of the starting codon of the *FAE1* gene, and this was observed exclusively in samples with a low erucic acid content [50].

Variations in the location of motifs in the *AOS2-like* promoter, particularly the endosperm-specific element (AACACOREOSGLUB1) associated with seed development [53], may influence changes in gene expression in seedlings. According to our data, the expression in roots of Leon seedlings was not recorded, while expression was observed in the Ratnik cultivar.

The promoter of the *PM19L-like* gene in the Leon cultivar did not contain the PRECONSCRHSP70A and MYBCORE presented in Ratnik. PRECONSCRHSP70A is a consensus motif presented in a plastid response element, known as a part of the *HSP70A* promoter of *Chlamydomonas* acting as an enhancer. *HSP70A* expression is upregulated through this motif by a chlorophyll precursor Mg-protoporphyrin and light [54]. In plants, the *HSP70* gene family is expressed in response to abiotic stresses. Plant transcription factors MYB binding to MYBCORE motifs can also activate the expression of genes involved in responses to abiotic stress, the promoters of which contain these motifs [52]. Significant upregulation of *PM19L-like* in the leaves of Ratnik seedlings after the γ-irradiation of seeds (Figure 4) may be associated with these motifs in promoters. It was shown that promoters of genes responding to abiotic stress, in particular, the *O. sativa PM19L* gene promoter, can be used to create stress-inducible promoters for genetic engineering. Therefore, it is possible to obtain overexpressors through promoter-region editing with increased tolerance to stressors without the reduction of plant growth and reproductive potential, in contrast to the use of constitutive promoters [31]. Thus, variations in barley promoter sequences can be used as starting points to create overexpressors of target genes.

## 4. Materials and Methods

### 4.1. Barley Cultivars

Seeds of eight barley cultivars (*Hordeum vulgare* L.) were used in the study: Ratnik, Grees, Fedos, and Leon (spring barley) and Fox 1, Eryoma, Vivat, and Timofey (winter barley). The seeds of all cultivars were provided by the Agrarian Science Center “Donskoy” (Zernograd, Russia), a harvest of 2020. Seed samples are available upon request.

In our previous work [21], these barley cultivars were ranked based on the morphological responses of seedlings to γ-irradiation of seeds at a dose of 20 Gy. Briefly, for ranking, we used four morphological indicators: (1) length of shoot, (2) length of root, (3) biomass of shoot, and (4) biomass of root of 7-day-old seedlings growing from irradiated seeds. All cultivars were subsequently divided into:-γ-stimulated—an increase in length and/or biomass (Fox 1, Ratnik, and Eryoma);-“no morphological effect”—no prominent changes compared with non-irradiated plants (Grees, Timofey, Vivat, and Fedos);-γ-inhibited—inhibition of growth compared with non-irradiated seedlings (Leon).

### 4.2. Irradiation of Seeds and Greenhouse Experiment 

Barley seeds were irradiated at a dose of 20 Gy (dose rate 60 Gy/h [18]) using γ-facility “GUR-120” (^60^Co) (RIRAE, Obninsk, Russia). Seeds were irradiated in small plastic envelopes (approximately 1000 seeds per envelope) at room temperature (20–22 °C). Non-irradiated seeds of each cultivar were used as a control.

Chernozem without inclusions was used for planting seeds. An agrochemical soil analysis is presented in Appendix A. The agrochemical properties were assessed according to the ISO standard for soil quality [55].

Seeds were planted in plastic growing containers with soil. Expanded clay was first poured onto the bottom of each container, completely covering the bottom. Two layers of gauze were put over the expanded clay. Then, 5.5 kg of dry soil was weighed on a balance STX8200 (OHAUS), and 600 mL of water was added to the soil and mixed in a basin until uniformly moist. The soil was carefully filled into labelled containers with expanded clay and gauze, containing an aeration glass tube. After filling each container with soil, its total mass was recorded. Thirty seeds were sown in each container, using a stencil and a dissecting needle or a glass pestle (0.5 cm in diameter) to deepen the seeds into the soil; the sowing depth was 2–2.5 cm.

A total of 48 containers were used, 6 for each cultivar (3 with control plants and 3 with irradiated plants). A total of 1440 seeds were used in the experiment. Plants were grown until ripe. The average temperature during the experiment was 25.5 °C, and humidity was 28.9%. During the experiment, plants were watered with distilled water, 150–200 mL, daily in each container.

### 4.3. Assessment of Developmental Stages

We monitored the change in developmental stages of plants throughout the entire ontogeny. To determine the stages of barley development, we used a scale [30] containing stages from 00 to 99. They were determined visually by phenological features of organ formation: germination (00–09); seedling growth (10–19); tillering (20–29); stem elongation (30–39); booting (40–49); inflorescence (ear/panicle) emergence, or earing (50–59); flowering (60–69); and three stages of grain ripening: milk grain (70–79), dough grain (80–89), and fully ripe (90–99).

For each container, the number of plants at each developmental stage was recorded. The stages were tracked from the moment of coleoptile emergence until the 40th DAP daily; from the 40th to the 57th DAP—every other day; from the 57th DAP to the end of the experiment—twice a week. 

The beginning/end of the ontogenetic stage was considered as a situation when at least one plant of the cultivar began/ended this stage. Data processing of the analysis of developmental stages was carried out using Microsoft Office Excel 2019.

### 4.4. Sampling for Gene Expression Assessment

Our previous screening studies revealed homologues of rice genes *CML31*, *AOS2*, and *PM19L* as promising candidates for radiation hormesis in barley [21]. To analyse their expression in a large greenhouse experiment through all ontogenetic stages, we sampled leaves and roots of barley cultivars at different stages of ontogenesis in the greenhouse experiment. 

At the seedling growth stage, we sampled roots and leaves; at the tillering, booting, and earing stages, we sampled leaves only (96 samples at the seedling growth stage, 30 samples at the booting, 18 samples at the tillering, and 43 samples at the earing stages). A total of 187 samples were sampled. Sampling details are given in Appendix A.

For sampling, 3 plants were carefully removed from each container using tweezers. Roots were washed in distilled water 2–3 times. At the seedling growth stage, roots and leaves of three plants were simultaneously cut off. Plant tissues were placed in cryovials (roots and leaves separately) and immediately frozen in liquid nitrogen. We cut off and analysed only leaf tissues for the other developmental stages.

### 4.5. Gene Expression Analysis

Frozen barley root tissue (up to 100 mg) was homogenised in liquid nitrogen with the addition of polyvinylpyrrolidone; then, the homogenate was used to isolate total RNA using the GeneJet Plant RNA Purification Kit (Thermo Fisher Scientific, Waltham, MA, USA) according to the manufacturer’s protocol. Frozen leaf tissue (up to 100 mg) was homogenised in liquid nitrogen with the addition of polyvinylpyrrolidone; then, the homogenate was used to isolate total RNA using a reagent ExtractRNA (Evrogen, Moscow, Russia) according to the manufacturer’s protocol. The quality of the isolated RNA was checked using a NanoDrop-2000 spectrophotometer.

In total, 1 μg of total RNA isolated from root and leaf tissues was subjected to treatment with DNase I (Thermo Fisher Scientific, Waltham, MA, USA) and subsequent cDNA synthesis using the MMLV RT kit (Evrogen, Moscow, Russia), according to the manufacturer’s instructions. cDNA was diluted 1:10 in nuclease-free H_2_O and used as a template for the real-time PCR.

Primers for *CML31-like*, *PM19L-like*, and *AOS2-like* were developed using Primer BLAST software (https://www.ncbi.nlm.nih.gov/tools/primer-blast/, accessed on 1 January 2019) [56]. The specificity of primer pairs was tested using qPCR and 2% agarose gel separation. Primer sequences, IDs, and functions of the encoded proteins are given in Appendix A.

Real-time PCR was performed using the qPCRmix-HS SYBR Green kit (Evrogen, Moscow, Russia) according to the manufacturer’s protocol. The 20 μL qRT-PCR reaction consisted of 4 μL of cDNA, 2 μL of the primer pair mixture (1 μM), 4 μL of qPCRmix-HS SYBR (Evrogen, Moscow, Russia), and 10 μL of nuclease-free H_2_O. The reaction was performed in a DT-96 amplifier (DNA Technology, Moscow, Russia) using the following conditions: preliminary denaturation for 5 min at 95 °C; then, 40 cycles of denaturation/annealing/elongation (15 s at 95 °C/15 s at 65 °C/60 s at 60 °C).

The actin gene (*ACT*) was used as a housekeeping reference. *ACT* is widely used as a reference gene in plant research [57] and was tested in our previous radiation-related experiments as stable after seed irradiation [11]. All analyses were performed in three biological and two technical replicates.

To calculate changes in gene expression, we used the ∆∆Cp model [58]. A significant change in gene expression was considered a twofold increase or decrease compared to the control.

### 4.6. Estimation of Yield

After plants reached the stage of seed ripeness, they were harvested, and the following parameters were assessed: number of steams, plant height, the average height of stems per plant, number of productive shoots, the weight of spikes, number of grains per spike, grain weight per spike, weight of 100 grains, and weight of straw [18,59]. The height of plants was measured using a ruler (systematic measurement error—0.1 cm). All weights were determined using analytical balance Pioneer (Ohaus, Parsippany, NJ, USA).

### 4.7. Estimation of Chlorophyll Fluorescence Parameters

Chlorophyll fluorescence was measured using a MultispeQ V 2.0 (PhotosynQ, East Lansing, MI, USA) on the 71st DAP when most barley plants were at the booting stage. Photosynthetic parameters were estimated for 3 plants per container using the same stage of leaf development for control and irradiated plants (9 measurements per condition, for a total of 144 plants).

Measurements were carried out on plants with a developing spike, using the last leaf before the flag leaf. In plants without spikes, chlorophyll fluorescence was measured in the last opened leaf of the main stem. The following parameters were taken into account: ECS—the electrochromic shift indicating the lifetime of steady-state proton translocation through the chloroplast ATP synthase; LEF—linear electron flow; NPQ—non-photochemical quenching; Phi2—quantum yield of photosystem II; PhiNO—a ratio of incoming light that is lost via non-regulated processes; PhiNPQ—a ratio of incoming light that goes towards non-photochemical quenching; g_H_^+^—steady-state rate of proton flux through the chloroplast ATP synthase; PS1—active photosystem I that is operational to receive/pass electrons; Fv/Fm—maximum quantum efficiency of photosystem II; qL—fraction of photosystem II centres which are in the open state; v_H_^+^—proton conductivity of the chloroplast ATP synthase [42,60,61,62].

### 4.8. Data Analysis of Yield and Chlorophyll Fluorescence 

Data analysis of yield and chlorophyll fluorescence was carried out using the Microsoft Office Excel 2019 software. The data were checked for outliers. Values that were 1.5 × IQR greater than the third quartile and 1.5 × IQR lower than the first quartile were designated as outliers and excluded from further analysis. Results are presented in “median (Q1; Q3)” format. The Mann–Whitney U-test in Statistica 12.0 was used to determine the significance of differences.

### 4.9. Analysis of Gene Structure of CML31-like, AOS2-like, and PM19L-like 

To sequence the barley homologues of *CML31*, *AOS2*, and *PM19L* rice genes, the latest annotation of the reference assembly from the EnsemblPlants database was used (Appendix A). Primers were designed using the Primer BLAST program (https://www.ncbi.nlm.nih.gov/tools/primer-blast/, accessed on 1 December 2022) [56]. Primers limited the length of sequenced fragments from 200 to 1000 bp with an overlap of 20–60% (Appendix A). The performance and specificity of the resulting primer pairs was tested on barley DNA using PCR and agarose gel visualisation. Up to 2000 bp of the promoter region of each gene was also considered for sequencing.

Genomic DNA was extracted from 3 individual barley seedlings of the cultivars Ratnik and Leon using the CTAB-based Sorb-GMO-B kit (Syntol, Moscow, Russia). DNA concentrations and quality were assessed on a NanoDrop OneC spectrophotometer (Thermo Fisher Scientific, Waltham, MA, USA) and diluted with nuclease-free water to a concentration of 20 ng/μL for PCR.

PCR was performed on DT-96 (DNA Technology, Moscow, Russia) in a 20 μL reaction mixture (4 μL of DNA, 2 μL of a 10 μM mixture of forward and reverse primers, 4 μL of the qPCRmix-HS reaction mixture (Evrogen LLC, Moscow, Russia), 10 μL nuclease-free H_2_O) under the following conditions: (1) 97 °C for 5 min, (2) 60 cycles at: 95 °C for 15 s, 62 °C for 30 s, 70 °C 1 min 30 s. When necessary, touchdown PCR protocol ((1) 97 °C for 5 min; (2) 15 cycles at 94 °C for 10 s, 67 °C for 20 s, 72 °C for 1 min 30 s; and (3) 35 cycles at 95 °C for 10 s, 60 °C for 20 s, 72 °C for 1 min 30 s) was applied. Two technical replicates were used for each sample. Electrophoresis was performed in a 1.5% agarose gel using SYBR Green and DNA length markers 100+ bp DNA Laddery (Evrogen, Moscow, Russia). Amplified fragments were excised from the gel, eluted using “ColGen” kit (Syntol, Moscow, Russia), and sequenced by Sanger at Syntol (Moscow, Russia). Data analysis and processing were carried out using Unipro UGENE v 45.1 (http://ugene.net/ru/, accessed on 1 March 2023) [63] and Clustal Omega 1.2.2 software [64]. The quality threshold for trimming the ends of the resulting sequences was set at 30, and the minimum similarity level for alignment was set at 80%. A strict consensus type for the reference assemblies of each gene was used during the alignment.

The cis-regulatory elements were searched using the electronic resource PlantCARE [65]. The search for transcription factor binding sites (TFBS) in promoter sequences 1.2 kb long located upstream of the target genes’ start codon was carried out using PLACE (http://www.dna.affrc.go.jp/PLACE/, accessed on 1 November 2023) [66].

## 5. Conclusions

Radiation hormesis can improve crop yield, immunity, and stress tolerance. However, precise molecular foundations of the effect remain elusive. In this work, we tested a set of previously revealed candidate genes of radiation hormesis in barley throughout ontogenesis in a greenhouse experiment. Coordinated analyses of growth dynamics and gene expression in irradiated and control plants confirmed that *PM19L-like*, *CML31-like*, and *AOS2-like* are plausible candidate genes of radiation hormesis, as their expression patterns are different between γ-stimulated and γ-inhibited barley cultivars and are coordinated with growth dynamics changes. The structure and regulatory *cis*-elements of target genes were analysed in two contrasting cultivars, showing that certain promotor elements may be involved in radiation hormesis effect execution. These data are necessary to create further genetically edited barley lines with improved yield and stress tolerance using Ratnik and Leon as parent cultivars.

## Figures and Tables

**Figure 1 ijms-25-00974-f001:**
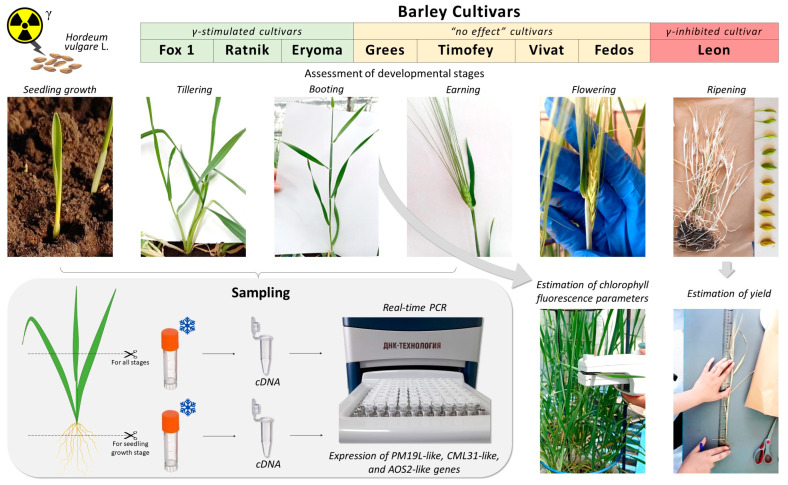
Examples of phenological stages of barley ontogenesis and a sampling scheme at different developmental stages.

**Figure 2 ijms-25-00974-f002:**
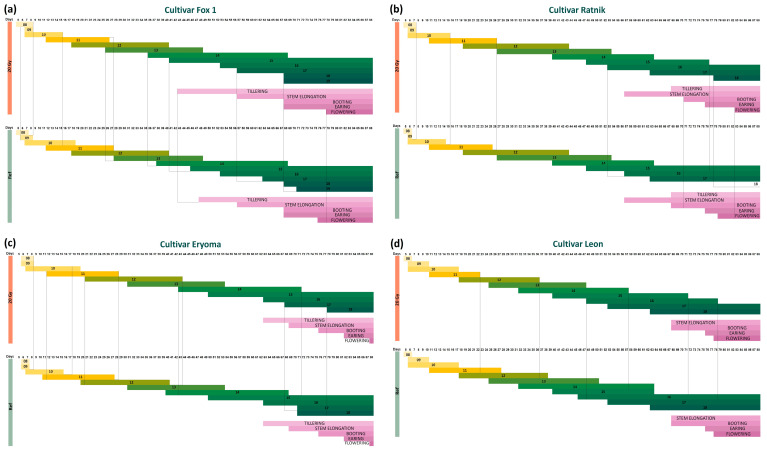
Dynamics of phenological stages of γ-stimulated cultivars: (**a**) Fox 1, (**b**) Ratnik, (**c**) Eryoma, and (**d**) γ- inhibited cultivar Leon.

**Figure 3 ijms-25-00974-f003:**
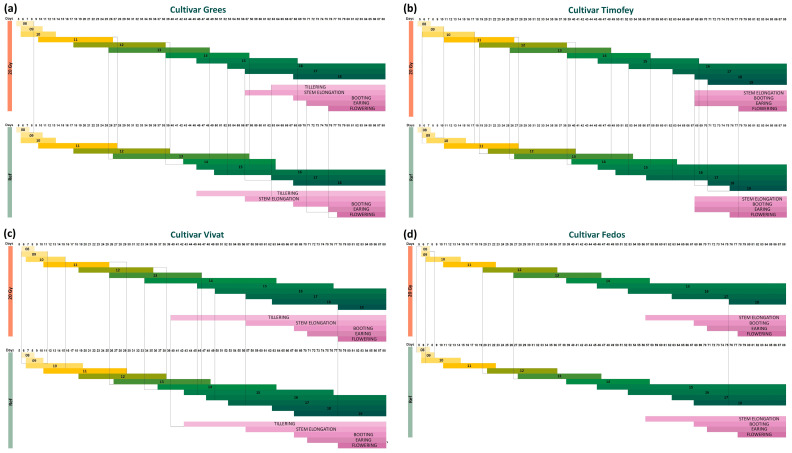
Dynamics of phenological stages of “no effect” cultivars: (**a**) Grees, (**b**) Timofey, (**c**) Vivat, and (**d**) Fedos.

**Figure 7 ijms-25-00974-f007:**
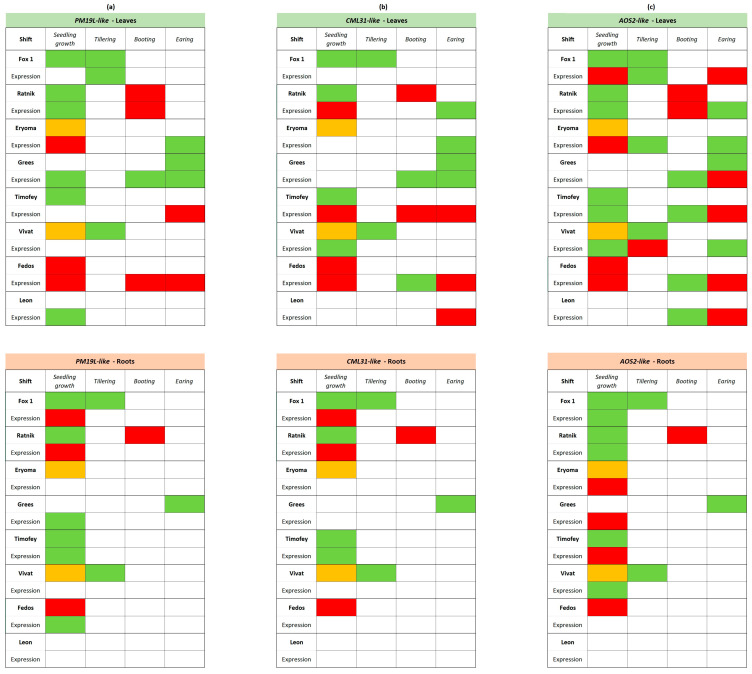
Summary of seed irradiation effects on growth dynamics and gene expression: (**a**) for *PM19L-like* in leaves and roots, (**b**) for *CML31-like* in leaves and roots, (**c**) for *AOS2-like* in leaves and roots. For each cultivar, the first line reflects changes in growth dynamics, and the second line—changes in gene expression at a certain growth stage. Red box—a gene is downregulated, or the stage began later in the irradiated plants than in the control. Green box—a gene is upregulated, or the stage began or ended earlier in the irradiated plants than in control; orange—some micro-stages started earlier, some later in irradiated plants than in control plants.

**Figure 8 ijms-25-00974-f008:**
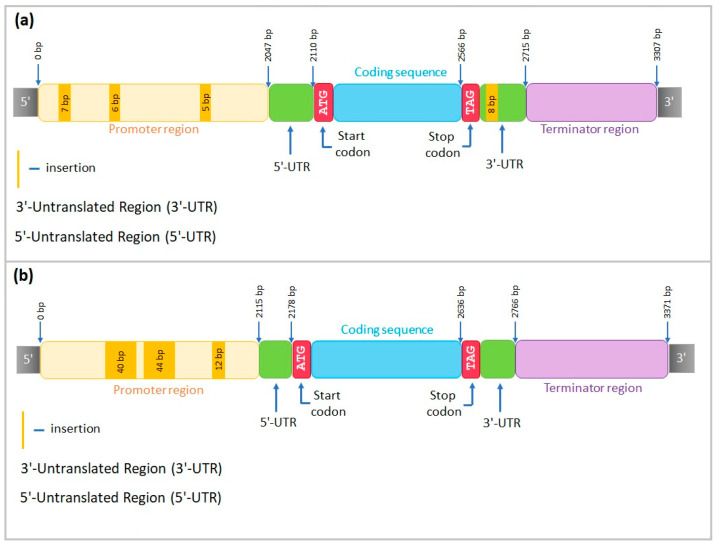
*CML31-like* structure in (**a**) Ratnik cultivar and (**b**) Leon cultivar.

**Figure 9 ijms-25-00974-f009:**
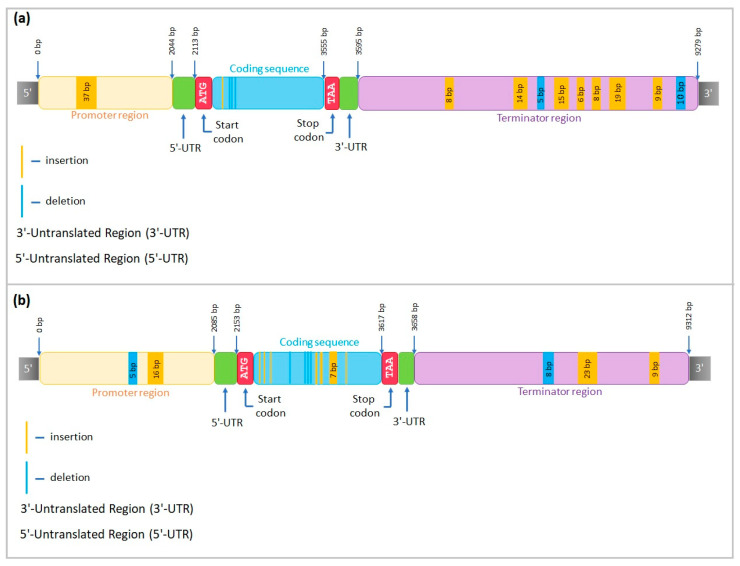
*AOS2-like* structure in (**a**) Ratnik cultivar and (**b**) Leon cultivar.

**Figure 10 ijms-25-00974-f010:**
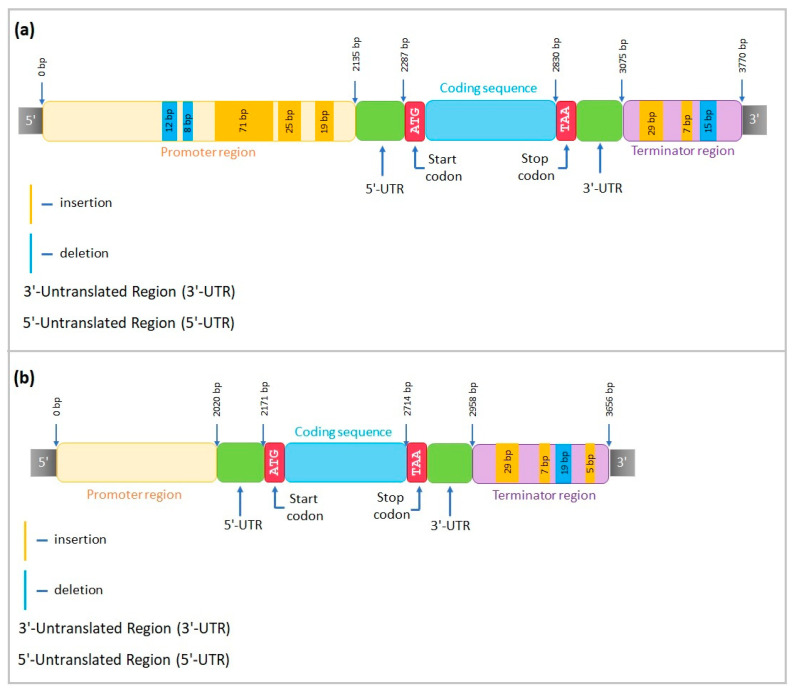
*PM19L-like* structure in (**a**) Ratnik cultivar and (**b**) Leon cultivar.

## Data Availability

Data are contained within the article and in Appendix A, and raw data are available on request from the corresponding author.

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
