# Peer review of "Radiation Hormesis in Barley Manifests as Changes in Growth Dynamics Coordinated with the Expression of PM19L-like, CML31-like, and AOS2-like"

_ijms, 2024, doi:10.3390/ijms25020974_

Round 1

Reviewer 1 Report

Comments and Suggestions for Authors

The manuscript “Radiation hormesis in barley manifests as changes in growth dynamics coordinated with the expression of PM19L-like, CML31-like, and AOS2-like” fits the scope of the section Molecular Plant Sciences. The authors evaluate the impact of low-dose γ-irradiation on eight barley cultivars, and showed the significance of previously identified genes (PM19L-like, CML31-like, AOS2-like) in radiation hormesis across ontogeny. The expression of these genes correlates with growth dynamics, yield parameters, and physiological conditions in γ-stimulated cultivars, providing insights for gene-editing. The study identifies cultivars with specific abiotic stress-responsive elements, potentially uncovering a new level of radiation hormesis. The manuscript is well written, the methods are described in suffiecient detail, and the presentation of the results is satisfactory. However, before publication the manuscript needs several corrections (please see below).

Section 4.1 – please specify if voucher specimens are available

Lines: 74-81 – please move the section to material and methods

Lines: 497-503 – please rephrase the paragraph for more clarity.

Lines: 596-597 – please check if the distribution of the results is not a consequence of the sampling method. Please clarify.

The conclusions should be more concise. Please shorten the section.

Please highlight the novelty of the study.

Please check if all self-citation are necessary.

Comments on the Quality of English Language

Please check the manuscript for minor editing/typing errors; Also please check the journal guideline in order to present correctly all reactives and apparatus.

Author Response

A point-by-point response to the Reviewers' comments

            The authors would like to thank three anonymous Reviewers for thoroughly checking the manuscript and for the valuable comments and suggestions. We considered all the points raised and provided a point-by-point response to each of them. All changes made in the manuscript are highlighted in the revised version.

Reviewer 1

The manuscript is well written, the methods are described in suffiecient detail, and the presentation of the results is satisfactory.

Dear Reviewer, thank you for the high estimate of our work and many important comments and suggestions that helped us improve the manuscript.

Section 4.1 – please specify if voucher specimens are available

We specified that seed samples are available upon request.

Lines: 74-81 – please move the section to material and methods

We moved this part to expand Section 4.3. Still, we did not delete this information from the results, as this allows readers not to "jump" to M&M immediately, understanding the concept of micro phenological stages used in the work.

Lines: 497-503 – please rephrase the paragraph for more clarity.

We rewrote Section 4.1

Lines: 596-597 – please check if the distribution of the results is not a consequence of the sampling method. Please clarify.

We are sorry that the description was somehow unclear. We added information that 9 plants were used per condition (either control or irradiated).

The conclusions should be more concise. Please shorten the section.

The conclusion has been rewritten.

Please highlight the novelty of the study.

The novelty has been highlighted in the Introduction part.

Please check if all self-citation are necessary.

Some unnecessary citations were removed, but most were retained as the current article describes the results of the almost 10-year search of radiation hormesis candidate genes, which involved many steps, including stimulating dose tests, cultivars choice, and omic analyses and validation.

Reviewer 2 Report

Comments and Suggestions for Authors

Review of "Radiation Hormesis in Barley: Insights into Growth Dynamics and Gene Expression": The manuscript authored by Elizaveta Kazakova, Irina Gorbatova, Anastasia Khanova, Ekaterina Shesterikova, Ivan Pishenin, Alexandr Armenovich Prazyan, Mikhail Podlutskii, Yana Blinova, Sofia Bitarishvili, Ekaterina Bondarenko, Alena Smirnova, Maria Lychenkova, Vladimir Bondarenko, Marina Korol, Daria Babina, Ekaterina Makarenko and Polina Volkova explores the effects of low-dose γ-irradiation on barley seeds, focusing on growth dynamics and the expression of specific genes—PM19L-like, CML31-like, and AOS2-like—across various developmental stages in eight barley cultivars.

The primary objective is to validate these genes' roles in radiation-induced growth stimulation and their potential as targets for gene editing.

The study meticulously investigates the correlation between the expression of the candidate genes and the growth patterns, yield parameters, and physiological conditions observed in γ-stimulated barley cultivars. The research convincingly establishes the significance of PM19L-like, CML31-like, and AOS2-like genes throughout the barley ontogeny, elucidating their crucial involvement in the growth stimulation induced by γ-irradiation.

Of particular note is the identification of specific barley cultivars exhibiting diverse responses to γ-stimulation, indicating promising avenues for future gene-editing strategies. Additionally, the manuscript unveils the presence of unique abiotic stress-responsive elements within the promotors of candidate genes in the γ-stimulated cultivar, hinting at a novel mechanism underlying radiation hormesis effects.

The implications of these findings extend beyond agriculture, encompassing ecological toxicology and eustress biology, offering valuable insights into the potential applications of radiation hormesis in crop development and stress response studies.

Overall, the manuscript significantly contributes to the comprehension of the genetic mechanisms governing radiation-induced growth stimulation in barley. It underscores the pivotal roles of PM19L-like, CML31-like, and AOS2-like genes as key players and potential targets for gene editing, promising advancements in enhancing crop resilience and productivity. The thoroughness of the research and its implications across diverse scientific fields render it a valuable addition to the literature on radiation hormesis and crop improvement strategies.

The research presented in this manuscript is interesting and valuable, but I have a few comments that the authors should take into account in the further process of preparing the manuscript for printing:

1) Please take into account that the Abstract section lacks a short research hypothesis and a clear outline of the practical application of the achievements presented in this manuscript.

2) Keywords should not repeat words included in the manuscript title. Additionally, keywords should be listed in alphabetical order.

3) The Introduction chapter provides a very poor background to the research topic. This section is definitely too brief and should be expanded.

4) The Conclusion chapter is very extensive and should be appropriately reworded and shortened. Please emphasize more the applied nature of the results obtained.

5) Figures 2 and 3 - the font size in the figures is too small, which limits their readability.

Finally, I would like to emphasize that the research results are interesting and very attractively presented, and the Discussion is substantive and based on a properly selected literature base. I believe that the Editors of the IJMS journal should consider publishing this manuscript.

Author Response

A point-by-point response to the Reviewers' comments

            The authors would like to thank three anonymous Reviewers for thoroughly checking the manuscript and for the valuable comments and suggestions. We considered all the points raised and provided a point-by-point response to each of them. All changes made in the manuscript are highlighted in the revised version.

Reviewer 2

Finally, I would like to emphasize that the research results are interesting and very attractively presented, and the Discussion is substantive and based on a properly selected literature base. I believe that the Editors of the IJMS journal should consider publishing this manuscript.

Dear Reviewer, thank you for the high estimate of our work and many important comments and suggestions that helped us improve the manuscript.

1) Please take into account that the Abstract section lacks a short research hypothesis and a clear outline of the practical application of the achievements presented in this manuscript.

Thank you, Abstract section was improved.

2) Keywords should not repeat words included in the manuscript title. Additionally, keywords should be listed in alphabetical order.

Thank you for attracting our attention to the issue; this has been resolved.

3) The Introduction chapter provides a very poor background to the research topic. This section is definitely too brief and should be expanded.

We intentionally kept the Introduction concise, as the Discussion is big. However, we followed the suggestion and expanded Introduction with a deeper discussion of hormesis effects and their implications.

4) The Conclusion chapter is very extensive and should be appropriately reworded and shortened. Please emphasize more the applied nature of the results obtained.

The Conslusion has been rewritten.

5) Figures 2 and 3 - the font size in the figures is too small, which limits their readability.

We increased the font of the Figures and provided high-resolution Figures which can be downloaded when the article is processed.

Reviewer 3 Report

Comments and Suggestions for Authors

Manuscript ID: ijms-2797858

Title: Radiation hormesis in barley manifests as changes in growth dynamics coordinated with the expression of PM19L-like, CML31-like, and AOS2-likeent carbon sources

 The submitted manuscript presents the results of research on the influence of low-dose γ irradiation on the dynamics of seed growth, the expression of selected genes, and the phenotypic features of tested barley varieties. The authors demonstrate that radiation hormesis can be utilized to obtain modified barley lines with increased yield and stress tolerance. In my opinion, this manuscript constitutes a valuable extension of existing knowledge on improving the efficiency of currently cultivated barley lines and it may be of interest to the readers of the journal. Overall, the article is well-written and organized. Below are some comments and suggestions that can further enhance the final version of the manuscript:

L. 75: It should probably be "00-99" instead of "08-99". The same applies to line 529, where "01-99" should be changed to "00-99".

l. 540: The phrase “large-scale experiment”, when used in the context of sampling at all stages of ontogeny, may be misleading. The authors likely intended to emphasize the comprehensiveness of the research. However, the wording used suggests that the research was conducted in field conditions over a large area, rather than on a laboratory scale.

Lines 149, 161, 531 and others: The authors consistently use the term 'germination' to describe all growth phases of the studied plants. However, it's worth noting that 'germination' typically refers to the initial phase of plant development. In this context, the authors appropriately use the term to describe the phase between 00-09 (l. 76).

Figures 2 and 3 are illegible, hindering a comprehensive interpretation. Furthermore, there is an inconsistency between the numbering of growth phases in the figures and the discussion on page 4, in contrast to the scale proposed in lines 75-79 on the same page.

Faster plant development does not necessarily mean a positive effect of the applied factor. The phenomenon of accelerated plant development in response to a negative stimulating factor is also known. Plants growing in unfavorable conditions usually shorten their vegetation period and expedite seed production (usually in smaller quantities) in the hope that the next generation will have chance to develop in a more favorable ecosystem. Taking this into account, the primary criterion for assessing the impact of radiation on barley should be the obtained yield. 

Author Response

A point-by-point response to the Reviewers' comments

            The authors would like to thank three anonymous Reviewers for thoroughly checking the manuscript and for the valuable comments and suggestions. We considered all the points raised and provided a point-by-point response to each of them. All changes made in the manuscript are highlighted in the revised version.

Reviewer 3

In my opinion, this manuscript constitutes a valuable extension of existing knowledge on improving the efficiency of currently cultivated barley lines and it may be of interest to the readers of the journal. Overall, the article is well-written and organized.

Dear Reviewer, thank you for the high estimate of our work and many important comments and suggestions that helped us improve the manuscript.

  1. 75: It should probably be "00-99" instead of "08-99". The same applies to line 529, where "01-99" should be changed to "00-99".

Thank you for pointing out this inconsistency; everything has been corrected.

  1. 540: The phrase "large-scale experiment", when used in the context of sampling at all stages of ontogeny, may be misleading. The authors likely intended to emphasize the comprehensiveness of the research. However, the wording used suggests that the research was conducted in field conditions over a large area, rather than on a laboratory scale.

Thank you, we agree with the comment and rephrased it accordingly to avoid misinterpretations.

Lines 149, 161, 531 and others: The authors consistently use the term 'germination' to describe all growth phases of the studied plants. However, it's worth noting that 'germination' typically refers to the initial phase of plant development. In this context, the authors appropriately use the term to describe the phase between 00-09 (l. 76).

Indeed, thank you – we changed it throughout the manuscript to "days after planting (DAP)".

Figures 2 and 3 are illegible, hindering a comprehensive interpretation. Furthermore, there is an inconsistency between the numbering of growth phases in the figures and the discussion on page 4, in contrast to the scale proposed in lines 75-79 on the same page.

We increased the font of the Figures and provided high-resolution Figures which can be downloaded when the article is processed. We checked the text for any inconsistencies and hopefully corrected all of them.

Faster plant development does not necessarily mean a positive effect of the applied factor. The phenomenon of accelerated plant development in response to a negative stimulating factor is also known. Plants growing in unfavorable conditions usually shorten their vegetation period and expedite seed production (usually in smaller quantities) in the hope that the next generation will have chance to develop in a more favorable ecosystem. Taking this into account, the primary criterion for assessing the impact of radiation on barley should be the obtained yield. 

We agree with the Reviewer, and this concern was the main reason we chose cultivar Ratnik over other stimulated cultivars (evident yield improvement). We added a sentence to the Discussion to make this point clearer.

On behalf of co-authors,

Dr. Polina Volkova

Dr. Elizaveta Kazakova